# Peripheral Immune Dysfunction: A Problem of Central Importance after Spinal Cord Injury

**DOI:** 10.3390/biology10090928

**Published:** 2021-09-17

**Authors:** Marisa A. Jeffries, Veronica J. Tom

**Affiliations:** Department of Neurobiology and Anatomy, Drexel University College of Medicine, Philadelphia, PA 19129, USA; maj359@drexel.edu

**Keywords:** autonomic dysreflexia, spinal cord injury, immune dysfunction, SCI-IDS

## Abstract

**Simple Summary:**

Spinal cord injury can result in an increased vulnerability to infections, but until recently the biological mechanisms behind this observation were not well defined. Immunosuppression and concurrent sustained peripheral inflammation after spinal cord injury have been observed in preclinical and clinical studies, now termed spinal cord injury-induced immune depression syndrome. Recent research indicates a key instigator of this immune dysfunction is altered sympathetic input to lymphoid organs, such as the spleen, resulting in a wide array of secondary effects that can, in turn, exacerbate immune pathology. In this review, we discuss what we know about immune dysfunction after spinal cord injury, why it occurs, and how we might treat it.

**Abstract:**

Individuals with spinal cord injuries (SCI) exhibit increased susceptibility to infection, with pneumonia consistently ranking as a leading cause of death. Despite this statistic, chronic inflammation and concurrent immune suppression have only recently begun to be explored mechanistically. Investigators have now identified numerous changes that occur in the peripheral immune system post-SCI, including splenic atrophy, reduced circulating lymphocytes, and impaired lymphocyte function. These effects stem from maladaptive changes in the spinal cord after injury, including plasticity within the spinal sympathetic reflex circuit that results in exaggerated sympathetic output in response to peripheral stimulation below injury level. Such pathological activity is particularly evident after a severe high-level injury above thoracic spinal cord segment 6, greatly increasing the risk of the development of sympathetic hyperreflexia and subsequent disrupted regulation of lymphoid organs. Encouragingly, studies have presented evidence for promising therapies, such as modulation of neuroimmune activity, to improve regulation of peripheral immune function. In this review, we summarize recent publications examining (1) how various immune functions and populations are affected, (2) mechanisms behind SCI-induced immune dysfunction, and (3) potential interventions to improve SCI individuals’ immunological function to strengthen resistance to potentially deadly infections.

## 1. Introduction

Spinal cord injury (SCI) is a traumatic injury that results in disrupted bidirectional communication between higher levels of the central nervous system (CNS) and the body below the level of the injury. While SCI is often associated with motor and sensory dysfunction, SCI results in a myriad of other systemic functional changes and deficits, including altered immune function. Immune dysfunction after SCI has long been documented. For instance, SCI individuals exhibit more frequent infections, with pneumonia ranking as a leading cause of death after injury [1]. Even when death is not the eventual outcome, pneumonia and post-operative wound infections are associated with impaired functional recovery in SCI persons [2]. While this increased susceptibility to infection was historically attributed to the medical interventions routinely administered in the acute phase after SCI [3], it is becoming apparent that the rise in infections is largely due to secondary changes in peripheral immunity that occur after injury [4]. Studies have only recently begun to reveal the underlying biological mechanisms of immune pathology after injury. Individuals with SCI display various immunological changes, including immunosuppression despite concurrent chronic systemic low-grade inflammation, termed SCI-induced immune depression syndrome (SCI-IDS) (Figure 1) [5,6]. Unfortunately, because of our limited understanding of mechanisms that contribute to SCI-IDS, there are no FDA-approved treatments for use in SCI individuals that specifically improve immune function. Therefore, SCI-IDS represents a problem with myriad remaining questions. Importantly, identification of potential therapeutic targets to improve immune function would make a significant and lasting impact on the general health and well-being of the SCI community. In this review, we summarize the current literature describing the chronic peripheral inflammation and increased susceptibility to infection characteristic of SCI-IDS, the mechanisms behind the development of immune dysfunction, and how these pathological changes might be ameliorated therapeutically.

## 2. The Consequences of Peripheral Immune Dysfunction

### 2.1. Chronic Low-Grade Inflammation

While much research has focused on the resulting neuroinflammation in the spinal cord after SCI, there is also persistent, low-grade, peripheral inflammation that has been identified in SCI individuals (Figure 1) [6,7]. Examination of serum cytokine levels from SCI persons revealed that proinflammatory cytokines IL-2, IL-6, tumor necrosis factor alpha (TNFα), and/or IL-1RA were significantly increased compared to their levels in able-bodied subjects [8,9]. Interestingly, SCI subjects who experienced pain, urinary tract infections (UTIs), or pressure ulcers displayed higher levels of these proinflammatory cytokines than those without [9]. Another study found that C-reactive protein (CRP), IL-2, and granulocyte macrophage colony stimulating factor (GM-CSF) were significantly increased while TNFα, IL-4, and granulocyte colony stimulating factor (G-CSF) were significantly decreased in SCI persons compared to controls [10]. In men with SCI regardless of injury level, blood serum concentration of CRP, IL-6, endothelin-1, and soluble vascular cell adhesion molecule (sVCAM) were all significantly increased, suggestive of chronic low-grade inflammation [11]. A recent whole-blood gene expression study found significantly upregulated Toll-like receptor signaling pathway genes in participants with chronic SCI compared to those without, supporting the presence of systemic inflammation [12]. Activation of circulating CD3^+^ and CD4^+^ T cells was increased after SCI, although CCR4^+^ HLA-DR^+^ regulatory T cells were concurrently expanded [12]. The exact roles of regulatory T cells are complex, and the functional implications of increased regulatory T cells are not fully understood. Moreover, it is not yet known if the increase in the CCR4+ HLA-DR+ regulatory T cells causes immune dysfunction or they expand in response to an altered immune environment. Nevertheless, these data reveal immune dysfunction after chronic SCI in humans, and the increased activation of T cells may contribute to long-term inflammation.

### 2.2. Increased Susceptibility to Infection

Despite a sustained state of inflammation in SCI persons, they experience increased susceptibility to pathogenic infection due to concurrent immunodeficiency (Figure 1). The significance of this is underscored by clinical data examining causes of death in SCI individuals [13,14,15,16]. The leading cause of death within the first year after a traumatic SCI in the Czech Republic was found to be pneumonia infection (17.1%), with UTIs making up 7.3% of all deaths [16]. Pneumonia remained the leading cause of death after a year in the SCI population at 14% of all deaths, with UTIs at 10.3%, pressure ulcers at 12.1%, and sepsis of unknown origin at 6.5%, making infections the leading cause of mortality after SCI at over 40% of all deaths [16]. In a 70-year study from Britain, the leading cause of death after SCI was respiratory (29.3%) with infections such as pneumonia making up the vast majority of these (23.5% of all deaths) [15]. Another 7.8% of deaths in this study were attributed to urinary tract infections and sepsis of unknown origin, further demonstrating the gravity of infections in SCI individuals [15].

Why are respiratory infections so prevalent and deadly in the SCI population? Preclinical and clinical studies indicate that injury level and severity contribute to infection susceptibility. Clinically, over 40% of deaths in the tetraplegic population of the long-term study in Britain were attributed to respiratory causes, including infection [15]. In a small study from Germany, pneumonia and influenza ranked in the top three causes of death for tetraplegics, but not paraplegics, and tetraplegic subjects had a significantly reduced life expectancy compared to paraplegics [17]. Part of the problem is that higher-level injuries result in loss of innervation to motor neurons that innervate the diaphragm and intercostal muscles, resulting in compromised respiratory motor control (reviewed in [18]). The lack of mobility in SCI persons also exacerbates this issue since exercise, like walking and running, has been shown to reduce pneumonia-related mortality [19,20,21,22]. However, what may make respiratory infections particularly deadly for SCI persons is that higher injuries above the level of thoracic segment 6 (T6) result in disruption of descending supraspinal input to sympathetic preganglionic neurons (SPNs) that innervate immune organs and modulate immune function, which we describe in more detail in a later section. This combination results in a heavily reduced capacity to effectively clear respiratory infections.

Preclinical studies in animal models have further increased our understanding of impaired immune responses and subsequent increased susceptibility to infection following SCI [23,24,25,26]. Due to immune impairment, mice with thoracic SCI exhibit reduced ability to clear influenza or mouse hepatitis virus (MHV) infection in the lungs or liver, respectively [23,26]. In fact, mortality rates in infected SCI mice were 40% after influenza infection and 100% after MHV infection. Specifically, these mice exhibited reduced numbers of influenza-specific CD8^+^ T cells or MHV-specific CD4^+^ T cells after viral infection [23,26]. In a follow-up influenza study, deficient CD8^+^ T cell infiltration and numbers were discovered to be mediated via corticosterone signaling, and administration of mifepristone to inhibit corticosterone throughout the experiment rescued numbers of virus-specific CD8^+^ T cells [27]. In another study, high thoracic (i.e., T3) hemisection in a mouse model of inducible bacterial pneumonia resulted in increased bacterial load in the lungs, indicative of inability to clear infection [25]. Moreover, rats with complete transection at T10 developed UTIs when inoculated transurethrally with a lower dose of *E. coli* than uninjured rats [24]. Histological analysis indicated that while inflammation in the bladder was virtually resolved by 14 days post-infection in uninjured controls, SCI rats displayed chronic inflammation of the bladder with mononuclear cell aggregates located within the lamina propria. Together, these studies demonstrate the independent risk of SCI on infection rates across a wide range of infection types, further underscoring the need for therapeutic advancement in treating SCI-IDS.

### 2.3. Effects of Immune Dysfunction on Other Physiological Processes

There are common secondary complications after SCI that are likely worsened by the chronic immune dysfunction observed in SCI persons (Figure 1). Osteoporosis is ubiquitous in the SCI population, resulting in acute rapid reduction in bone density after injury that stabilizes after 2–3 years; this bone loss also increases susceptibility to fractures [28,29]. While the role of immune function in the pathogenesis of osteoporosis in the SCI population is not well described, osteoporosis independent of SCI is an inflammatory condition that progresses due to immune dysfunction, cytokine release, and a persistent low-grade inflammatory state typically seen in aging [30,31,32]. Therefore, it is not unreasonable to suggest that the long-term inflammation observed in SCI individuals would contribute to osteoporosis pathology [33].

Another secondary complication after SCI is neuropathic pain, which presents in anywhere from 18–96% of SCI persons [34,35,36,37,38,39]. As with osteoporosis, it is well established that both peripheral and central inflammation contribute to the development of neuropathic pain [40,41,42]. As described above, one study found that SCI persons presenting with neuropathic pain exhibited elevated serum levels of IL-6 and IL-1RA compared to those without neuropathic pain [9]. A recent clinical study indicated that an anti-inflammatory diet in SCI individuals resulted in reduced composite score of proinflammatory mediators IL-2, IL-6, IL-1β, TNF-α, and interferon gamma (IFN-γ) and was associated with decreased neuropathic pain score [43].

SCI individuals are frequently plagued by pressure ulcers due to immobility and resultant tissue ischemia. Data indicate that skin ulcers and cutaneous wounds heal more slowly after thoracic SCI in mice [44]. Cutaneous wounds normally progress through four stages of healing: hemostasis, inflammation and cytokine release, cytokine-induced epithelial and vascular proliferation, and wound resolution [45]. In the general population, wounds such as pressure ulcers often persist chronically in an inflammatory state that inhibits healing progression [45,46]. In line with this, clinical evidence suggests that anti-inflammatory topical treatments, such as TNFα inhibition via infliximab, can improve wound healing of chronic ulcers [47]. However, SCI persons exhibit sustained baseline vasodilation due to sympathetic denervation. This sustained baseline vasodilation and subsequent hypotension has been hypothesized to impair the requisite inflammation phase of wound healing, inhibiting wound healing at an earlier stage of repair [48]. In fact, evidence specifically from SCI models has shown that cutaneous inflammatory stimulation does not elicit appropriate localized inflammation. After complete T3 transection, mice injected subcutaneously with complete Freud’s adjuvant emulsion exhibited reduced cutaneous localized inflammation as measured by both fluorescent IVIS imaging and magnetic resonance imaging (MRI) [49]. Therefore, it seems that both persistent low-grade inflammation and immunosuppression after SCI may impair pressure ulcers from recovery.

The general consensus is that SCI increases the risk of atherosclerotic diseases, such as coronary artery disease and cardiovascular disease, via a multitude of secondary complications, from obesity to metabolic syndrome and sustained, low-grade inflammation [11,50,51,52,53,54]. While recent clinical research indicates that presentation of atherosclerotic pathology in SCI persons is not solely dependent on increased inflammatory markers [55,56], it may be exacerbated by increased inflammation [56]. Preclinical research using a mouse model of atherosclerotic disease (*ApoE^−/−^*) found that atherosclerotic lesions were significantly increased in mice with a T9 contusion injury compared to uninjured controls [57]. While the development of atherosclerosis was associated with increased plasma levels of IL-1β, TNFα, IL-6, monocyte chemoattractant protein-1 (MCP-1), and C-C motif chemokine ligand-5 (CCL-5), increased MCP-1 and CCL-5 were specifically observed in SCI mice versus uninjured controls. Importantly, the use of an anti-inflammatory salicylate drug was found to prevent SCI-induced exacerbation of atherosclerosis, possibly via the reduction in TNFα, MCP-1, and CCL-5 plasma levels [57].

Research therefore indicates that immune dysfunction in SCI individuals is of pathological consequence and resolution of chronic inflammation and concurrent immunosuppression could prove highly beneficial in improving quality of life.

## 3. Why Does Peripheral Immune Dysfunction Occur?

### 3.1. Disruption of Descending Central Pathways

SCI, particularly above the level of T6, can result in loss of modulatory input to immune organs via autonomic innervation and the hypothalamic-pituitary-adrenal (HPA) axis (Figure 2). The spleen, the largest lymphoid organ, has been better studied in relation to SCI-IDS than any other lymphoid tissue and shows dramatic changes after disruption of modulatory innervation [58]. Upon SPN activation, which normally occurs due to stress in the “fight or flight” response, post-ganglionic terminals in the spleen release norepinephrine (NE) directly to splenocytes in the white pulp and the HPA axis is stimulated to release glucocorticoids (GCs) from the adrenal gland (Figure 2) [59,60]. Under homeostatic conditions, splenic lymphocytes express anti-inflammatory β-adrenergic receptors (β-AR) that promote reduced cell proliferation, decreased proinflammatory cytokine release, and reduced antibody production. GCs from the adrenal glands mediate similar anti-inflammatory effects via GC receptors [61]. During inflammation, T cells and B cells highly express α-adrenergic receptors (α-AR) that promote maturation, activation, and migration (reviewed in [58,62]).

After SCI, supraspinal control of the sympathetic system is disrupted and sympathetic activity becomes dysregulated. Acutely, this results in increased GC release from the adrenal glands that impairs immune function (Figure 2) [63]. Over time, maladaptive plasticity of the sympathetic circuitry within the spinal cord below the level of injury develops. In people with severe high-level SCI, the combination of interruption of supraspinal input to sympathetic circuitry caudal to the level of injury and the plasticity of this circuit leads to sympathetic hyperreflexia, which overtly manifests as autonomic dysreflexia (AD) [64,65,66]. This sympathetic hyperactivity results in abnormally high levels of NE in the spleen and activation of β-ARs on lymphocytes that result in sustained immunosuppression (Figure 2) [67,68]. In turn, this chronic immunosuppression increases susceptibility to infection as described above. Sympathetic innervation and the HPA axis have been identified as both independent and interrelated causes of immune dysfunction after SCI [63,64].

### 3.2. Sympathetic Hyperreflexia

Recent studies have determined that sympathetic hyperreflexia contributes to SCI-IDS. Moreover, because sympathetic hyperreflexia development is correlated with injury level and severity, the extent and nature of immune dysfunction after SCI is injury level and severity dependent [25,58,63,64,65,67,69]. For instance, in the human SCI population, tetraplegic individuals display exacerbated increases in levels of the proinflammatory marker CRP compared to paraplegic individuals [70]. While chronic SCI persons exhibit fewer circulating CD3^+^ and CD4^+^ T cells and increased activation of remaining T cells, increased activation is particularly evident in those with complete or high level (above T6) injuries [71]. One preclinical study using rats showed that both pro-inflammatory and anti-inflammatory markers in plasma were significantly higher following a clip compression injury at T6/7 than at cervical level 6/7 (C6/7) [69]. It was suggested that this difference was due to the development of SCI-IDS in the cervically injured rats, though this was not directly demonstrated. Similarly, mice displayed splenic atrophy and leucopenia after a complete transection injury at T3 but not at T9 [64,65]. By 5 weeks post-injury, 50–70% of leukocytes were depleted, with a 60% reduction in the splenic B cell population. Moreover, T3-transected mice produced significantly lower antibody titers than uninjured controls when immunized with ovalbumin (OVA) antigen. These changes coincided with the development of AD [64]. Splenic white pulp atrophy and loss of B cells was exacerbated in the T3 SCI animals when sympathetic hyperreflexia was experimentally elicited with noxious sensory stimuli, such as colorectal distension (CRD). CRD also worsened the impaired immunological response after OVA antigen inoculation. Underscoring this, there was no significant effect on splenic B cells in the T9 injured group, which did not experience sympathetic hyperreflexia, indicating that sympathetic hyperreflexia is a causative factor in the development of SCI-IDS (Figure 2) [64].

Increased activation of vesicular glutamate transporter 2 (VGLUT2)+ excitatory interneurons in the lateral horn of thoracic spinal cord after injury has been implicated in the development of sympathetic hyperreflexia. These glutamatergic interneurons synapse on SPNs, and the number of presynaptic puncta contacting SPNs increases with time after SCI [65]. Chemogenetics, specifically designer receptors exclusively activated by designer drugs (DREADD), have also been used preclinically with great success to effectively determine the role of these excitatory interneurons in the intermediate and medial grey matter of the thoracic spinal cord after injury [65]. In this study, the researchers performed a T3 SCI in adult mice that expressed Gi/o-coupled human muscarinic M4 (hM4Di DREADD) in VGLUT2+ interneurons within the thoracic spinal cord. Clozapine-N-oxide (CNO) was injected starting two weeks after injury to silence the hM4Di-expressing, excitatory interneurons in thoracic spinal cord. Mice with this treatment exhibited normal splenic size and numbers of CD4^+^ T cells, CD8^+^ T cells, and B220^+^ B cells [65]. While this study did not directly examine whether these immune changes were functionally relevant, it revealed the causative role of maladaptive neural plasticity in immunological changes after SCI and indicated that targeting glutamatergic interneurons specifically may be a promising therapeutic target to ameliorate immune dysfunction.

Concurrent increases in blood endogenous GCs and splenic NE levels also appear to play a role in SCI-IDS (Figure 2). Systemic coadministration of selective antagonists for β-2 adrenergic and GC receptors resulted in reduced splenic atrophy and normal antibody titer after OVA immunization, suggesting the importance of sympathetic control in regulating immune function [64]. Importantly, severing the splenic nerve to obliterate sympathetic innervation prior to T3 SCI in mice abrogated the increased susceptibility to pneumonia infection normally observed in animals after T3 SCI with intact sympathetic signaling [25]. These studies indicate that the severity of immune dysfunction is strongly tied to sympathetic activity.

A notable caveat of preclinical studies using complete SCI models to elicit SCI-IDS is whether this complete loss of supraspinal input to the SPNs is recapitulated in humans with SCI. This is highly relevant because human injuries classified as clinically complete often are anatomically incomplete and have some tissue sparing [72,73,74,75,76]. So then, what happens to immune function after incomplete SCI, particularly below the level of T6? Preclinical studies found that splenic atrophy did not occur in rats with moderate, incomplete injuries at either cervical (C6/7) or thoracic (T6/7) levels [69,77]. However, splenic NE, corticosterone, and leukopenia were significantly increased within a week after the incomplete thoracic injuries but not cervical injuries [77]. Additionally, circulating pro- and anti-inflammatory cytokines and chemokines were increased in thoracic-injured rats compared to those with cervical injuries [69]. In line with this, some preclinical studies already described in this review reported changes in immune profile and function after incomplete SCI, even below T6 [26,27,78]. One factor that may contribute to this is that some of the SPNs that modulate sympathetic input to the spleen are located within T6/T7 spinal cord and are likely directly injured by even a moderate, mid-level SCI.

Our personal observations further support the concept of SCI-induced immune changes in the absence of overt splenic atrophy. While we found measurable splenic atrophy at 8 weeks post-complete T3 transection in rats, we observed that the splenic immune cell profile was already significantly altered by 4 weeks post-complete T3 SCI, though the spleens grossly appeared similar at that point (unpublished personal observations; [79,80]).

As described in earlier sections, one clinical study found that proinflammatory markers, such as CRP, were increased after SCI regardless of injury level [11], while another study found that the severity of increased CRP correlated with injury level [70]. Similarly, while activation of T cells was significantly increased in the general SCI population, individuals with complete or high-level injuries above T6 displayed a more pronounced effect [71]. What is not clear from these aforementioned studies is how injuries at different levels and severities affect sympathetic tone and if this plays a role in the disparate results described. It appears that injury level and severity contribute to the extent of immune dysfunction after SCI, but this is not absolute and the level dependence of SCI-induced immune dysfunction is likely complex.

### 3.3. Aberrant Activity of the HPA Axis

One recent study found that dysregulated HPA axis function after sympathetic disruption corresponds with more severe acute leucopenia after high thoracic injury. Specifically, mice with a T1 complete transection displayed acute reduction in systemic NE and increase in plasma GCs while those with a T9 complete transection did not [63]. This increase in GCs was due to adrenal gland denervation after the high thoracic complete injury resulted in aberrant hypercortisolism. In line with this, T1-transected mice displayed reduced numbers of CD19^+^ B cells, CD4^+^ or CD8^+^ T cells, CD11b^+^ monocytes, NK1.1^+^ natural killer (NK) cells, and CD11c^+^ dendritic cells (DC) in multiple lymphoid organs, including the spleen (Figure 2).

### 3.4. Disrupted Bone Marrow Function

Bone marrow is a key hematopoietic organ where bone marrow hematopoietic stem cells reside that give rise to myeloid and lymphoid cells, replenishing immune cell populations daily. Sympathetic innervation to the bone marrow regulates both bone turnover and immune cell production (reviewed in [81,82]). It is worth mentioning that the ubiquitous osteoporosis experienced by the SCI population is exacerbated by sympathetic hyperreflexia, resulting in reduced bone production and increased bone resorption [83,84,85]. These changes in bone turnover are part of an interconnected loop in which bone denervation promotes osteoporosis and immune dysfunction, which in turn bidirectionally affect each other.

Just how exactly bone marrow-derived immunity changes after SCI has been described in a few publications [86,87,88]. Clinically, persons with SCI exhibit impaired bone marrow stem cell function [86,87]. In particular, SCI individuals displayed impaired NK cytolytic function, reduced T cell killer function, and lower IgG levels indicative of inhibited B cell function despite normal circulating lymphocyte numbers. When bone marrow aspirates were cultured, the number of long-term culture-initiating cells was significantly reduced in cultures from SCI persons, particularly tetraplegics, indicative of decreased progenitor growth [86]. Preclinically, a recent publication explored the mechanisms behind SCI-induced bone marrow hematopoietic dysfunction [88]. After SCI, mice exhibited increased hematopoietic stem cell proliferation and accumulation in the bone marrow, as well as impaired mobilization regardless of injury level and severity (Figure 2). In T3-transected mice specifically, expression of bone marrow cytokines and chemokines was significantly increased, and C-X-C motif chemokine ligand 12 (CXCL12)/C-X-C motif chemokine receptor 4 (CXCR4) signaling specifically led to sequestration of hematopoietic stem cells and mature B cells. These changes appear to have functional implications, as the bone marrow response to inflammatory stimulation with lipopolysaccharide (LPS) was impaired after SCI [88].

### 3.5. Obesity

Several aforementioned studies observed increased susceptibility to various infections in rodents with lower thoracic injuries at T9/10, which largely leave descending control of sympathetic circuitry intact [23]. Therefore, while disruption of sympathetic innervation to lymphoid organs strongly contributes to immune dysfunction after SCI, it is not the only cause. Although the primary insult may be in the spinal cord, SCI is an injury of nearly every system in the body, from gastrointestinal to cardiovascular. The multi-system dysfunction observed after SCI results in a clinical population more likely to suffer from complications such as obesity and type 2 diabetes [89,90,91]. Indeed, while sympathetic hyperreflexia and subsequent AD appear to be a major underlying cause of SCI-IDS, concomitant chronic low-grade inflammation has been strongly linked to neurogenic obesity (reviewed in [92]) as well as other secondary complications of SCI (reviewed in [6,93]). Importantly, these conditions also can contribute to the development of immune dysfunction after SCI.

Obesity, which affects approximately 66% of SCI individuals [94], is thought to be a primary cause of the chronic low-grade inflammation observed after SCI [92]. Adipocytes have been shown to release “adipokines” such as TNFα, IL-6, and MCP-1, resulting in a systemic proinflammatory state in obesity [92,95]. In the SCI population specifically, higher waist circumference is associated with elevated CRP, a proinflammatory cytokine implicated in cardiovascular disease [70,96]. Evidence indicates that exercise to mitigate obesity can reduce systemic inflammation [97,98]. In SCI individuals specifically, plasma levels of proinflammatory cytokines TNFα and IL-6 were reduced after an arm cranking exercise regimen that improved anthropometric index, decreased waist circumference, and decreased plasma concentration of leptin [99]. Similarly, 10 weeks of functional electrical stimulation cycling by SCI persons resulted in increased muscle mass by dual X-ray absorptiometry and significantly reduced levels of proinflammatory cytokines IL-6, TNFα, and CRP [100].

Obesity, in turn, increases the risk of developing type 2 diabetes [101], a condition which is strongly associated with persistent systemic inflammation (reviewed in [102]). Studies have shown that SCI individuals are at higher risk of developing type 2 diabetes [89,91,103]. Interestingly, type 2 diabetes is considered to be immune-driven yet also contributes to immunosuppression via diabetes-mediated hyperglycemia [104,105]. While the role of type 2 diabetes in immune dysfunction specifically in the SCI population has yet to be established, it is highly possible that the known effects of insulin resistance and hyperglycemia on immunity carry over. These secondary complications arising from SCI therefore provide some explanation as to why SCI persons experience chronic low-grade systemic inflammation.

### 3.6. Repetitive Infections and Wounds

Persistent bacterial infections are thought to manipulate the immune system to prevent clearance [106,107]. In SCI individuals, repetitive infections, namely UTIs and infected chronic pressure ulcers, may contribute to both systemic low-grade inflammation and concurrent immunosuppression. As described in an earlier section, when SCI persons present with UTIs or pressure ulcers, serum levels of proinflammatory cytokines such as IL-6 and TNFα are significantly higher than in SCI persons without these infections [9]. This would suggest that ongoing infection can exacerbate the systemic inflammation observed after SCI. Along with this, SCI persons with an ongoing UTI displayed higher levels of urine IgA concentrations compared to those without infection, and SCI individuals displayed sustained IgG response to bacterial antigens despite no differences in circulating T cells specific to UTI bacterial antigens, compared to controls [10]. Interestingly, AD events have been documented to result in reduced oxygenation and increased perspiration of the skin, which in turn may contribute to increased susceptibility to pressure ulcers [108]. In turn, pressure ulcers and UTIs can both serve as stimuli that elicit AD events, which can further impair immune function after SCI, as described above. While it is still unclear to what degree persistent UTIs and pressure ulcer infections modulate immune activity in SCI individuals, it is apparent that an ongoing infection correlates with additional immunological changes.

## 4. Potential Interventions to Improve Immunological Function Post-SCI

### 4.1. A Critical Need for Clinical Therapies

There are no currently approved therapies specifically targeting improving the immune system for SCI individuals. In fact, the routine use of methylprednisolone in acute treatment of SCI persons in the United States is of debatable benefit for various reasons (reviewed in [109,110]), including the known effect of immunosuppression. While some studies have suggested that motor function recovery may be incrementally improved with methylprednisolone treatment, others have not found measurable effects but did note appreciable side effects [111,112]. Therefore, whether using methylprednisolone is of benefit is highly debated, even more so now that attention has turned to its immunosuppressive properties as a corticosteroid [113]. Another standard treatment for SCI individuals is rehabilitative medicine. With respect to systemic inflammation specifically, several studies have found that exercise lowers circulating levels of TNFα, IL-6, and CRP [99,114].

One method to improve quality of life in SCI individuals would be to reduce likelihood of infection. Clinical studies have examined the prophylactic use of antibiotics to prevent UTIs in the SCI population [14,115]. However, while several groups found that prophylactic low-dose clindamycin, sulfamethoxazole, nitrofurantoin, trimethoprim, or cefalexin treatment was effective in significantly reducing the rate of UTIs during extended treatment [116,117,118], another group found that prophylactic sulfamethoxazole or nitrofurantoin was not effective in reducing UTIs in the SCI population [119]. Additionally, antibiotic-resistant bacterial colonization of the bladder is common with prophylactic use of antibiotics in SCI persons [118,120], suggesting that prevention of infections via this route is unlikely to prove beneficial in the long term. To circumvent this problem, two studies attempted long-term prophylactic use of four antibiotics on a cyclical regimen; this method resulted in fewer yearly UTIs in SCI individuals [121,122]. Prophylactic antibiotic treatment has also been recommended by the North American Spine Society in cases of spinal surgery [123], with clinical studies often, but not always, indicating a reduction in post-operative infections, particularly with multiple days of treatment [124,125]. How these treatments might affect other types of infections in SCI persons remains unknown. Additionally, how recurring use of antibiotics affects the SCI body’s gut microbiome (discussed in more detail below) and the downstream consequences of that is not well understood.

Alternatively, therapeutic options could address underlying causes of immune dysfunction. As mentioned above, one cause of SCI-IDS is sympathetic hyperreflexia. There have been multiple preclinical and clinical studies using procedures and pharmacological interventions to both limit the onset of AD and manage it (reviewed in [126,127]). The current medical advice is that persons presenting with AD manage their symptoms using nonpharmacological methods, such as removing the offending stimulus (i.e., blocked catheter, tight clothing, or bowel impaction) and moving to a sitting position [128]. While these methods are typically successful in eventually ending the AD event, sustained high blood pressure can occur that demands further medical attention [66,128]. Commonly used drugs for the treatment of AD solely mitigate hypertensive symptoms [127], and therefore have short-lived effects that do not cure the underlying disorder. These drugs include nifedipine and nitrates, amongst other hypertension medications, such as angiotensin I converting enzyme inhibitors. Epidural stimulation has also been used to stabilize variable blood pressure in SCI persons [129]. Interestingly, preclinical work has revealed mixed data on the use of exercise to mitigate AD. While one study found that passive hindlimb cycling or active forelimb swimming did not change AD severity after T2 contusion injury in rats [130], another study found that passive hindlimb cycling after complete T3 transection did reduce the severity of AD [131]. Other studies have used botulinum toxin, capsaicin, anticholinergics, or surgical procedures to prevent the development or continuation of AD [126]. These treatments have had variable success in reduction in AD events in SCI persons [126,132,133]. However, it is important to keep in mind that sympathetic hyperreflexia affects not only the vasculature but also any organ that receives sympathetic innervation. None of these studies have examined effects on immune function, regardless of observed changes in AD presentation. It is possible that examining immune function after such treatments would reveal immunological changes, which would be of interest given the importance of improving immunity post-SCI. Additionally, there are no approved treatments to prevent the development of sympathetic hyperreflexia from the outset. Therefore, there is a dire need for the identification of treatments that improve immune function after SCI. In the following sections, we summarize recent preclinical research examining potential means to specifically improve immune function after SCI.

### 4.2. Gabapentin

Some preclinical studies have attempted to reduce the maladaptive neural plasticity that contributes to AD worsening via pharmacological intervention. Gabapentin (GBP), an anti-seizure and neuropathic pain medication known to prevent synaptogenesis at high doses, has been used after SCI in preclinical models to examine its effect on AD. Several studies have indicated that acute treatment with a low-dose of GBP (50 mg/kg) or chronic treatment with a very high-dose of GBP (400 mg/kg/day) starting the day of complete T4 SCI in rats decreased mean arterial pressure in response to CRD [134,135,136,137]. However, chronic treatment with this very high-dose of GBP (400 mg/kg/day) also increased the frequency of spontaneous AD events [136]. On the other hand, in another recent study, chronic treatment with a slightly lower dose of GBP (200 mg/kg/day) starting one day after complete T4 SCI in mice prevented excitatory synaptic formation and sprouting of sensory afferents, two examples of spinal plasticity associated with sympathetic hyperreflexia. This resulted in reduced frequency of spontaneous AD events, attenuated induced AD by CRD, and, importantly, mitigated changes in immune profile after SCI (Figure 3) [137]. Additionally, chronic treatment with this dose of GBP resulted in prevention of splenic atrophy and maintenance of CD3^+^ T cell and B220^+^ B cell populations in the spleen (Figure 3) [137]. One important difference in these seemingly conflicting studies is that Eldahan et al. did not observe any changes in excitatory or inhibitory presynaptic markers in the lumbosacral dorsal horn at the very high dose of GBP. Species/strain dependent differences may also account for the divergent results. While studies have presented conflicting data on the use of GBP for prevention of AD, the data from Brennan et al. support the notion that suppression of neural plasticity in the sympathetic circuit below the level of injury can improve the immune profile after SCI.

### 4.3. Inhibiting TNFa

Other studies have targeted TNFα signaling to improve sympathetic hyperreflexia and splenic function. TNFα expression is persistently upregulated in the spinal cord after injury, contributing to neural plasticity. TNFα exists in two forms; its soluble form is a product of transmembrane TNFα cleavage by TNFα-converting enzyme. While the former is highly proinflammatory and plays a role in neural plasticity below the level of injury [138], the latter has been shown to have neuroprotective effects [139,140]. After SCI, specific central inhibition of soluble TNFα using the experimental compound XPro1595, which inhibits soluble TNFα specifically, resulted in improved functional recovery [141]. Importantly, these effects were not replicated with central administration of etanercept, a pan-TNFα inhibitor that affects both soluble and transmembrane TNFα [141]. This highlights the particular role of soluble TNFα in SCI pathology, and paved the way for examination of how soluble TNFα inhibition after SCI might affect other facets of recovery, including immune function.

We recently reported that continuous intrathecal administration of XPro1595 in rats with complete T3 transections beginning up to 3 days post-injury significantly reduced intraspinal plasticity within the sympathetic circuit and lowered the frequency and severity of AD events (Figure 3) [79,80]. Additionally, in XPro1595-treated rats, splenic atrophy was prevented, and splenic immune cell profile was similar to non-injured control spleens, in contrast to spleens from injured animals without XPro1595 that exhibited an altered immune cell profile. In particular, CD45R^+^ B cells, CD8^+^ T cells, and CD11b/c^+^ macrophages were returned to uninjured numbers, while CD4^+^CD25^+^FoxP3^+^ regulatory T cells were significantly increased (Figure 3) [79,80]. Improved splenic immune profile corresponded with reduced sympathetic, noradrenergic sprouting in the spleen [79]. Excitingly, the improved immune profile in turn resulted in reduced susceptibility to pneumonia infection, with no XPro1595-treated injured rats dying while nearly 40% of vehicle-treated injured rats succumbed to infection. While vehicle-treated injured rats that survived exhibited persistent weight loss at 10 days post-infection, those that received XPro1595 returned to baseline [79]. However, when XPro1595 treatment was delayed until 2 weeks post-injury, no beneficial effects on AD were observed, suggesting that administration at some point prior to 2 weeks is vital to the effectiveness of this particular treatment strategy [142]. Nevertheless, the benefit of central soluble TNFα inhibition after subacute injury to attenuate sympathetic hyperreflexia and to improve downstream immune function is particularly promising given the striking increase in resistance to pneumonia infection.

### 4.4. Modulation of Gut Microbiota

Another possible therapeutic option may exist in gut microbiota. Notably, ~70% of immune cells reside in gut-associated lymphoid tissues (GALTs), where they respond to microbial antigens and metabolites produced in the gut and serve as the first line of defense against pathogens entering via the gastrointestinal route [143]. The links between gut microbiota, neurological function, and immunological pathology are research topics of great interest in diseases ranging from multiple sclerosis to autism [144,145,146,147], and the downstream effects of SCI on this complex system are only now beginning to be revealed.

In a landmark study examining microbiota changes after SCI, a T9 contusive injury in mice resulted in immune disruption, e.g., altered numbers of B220^+^ B cells, CD4^+^ and CD8^+^ T cells, CD11b^+^ macrophages, and CD11c^+^ DCs in GALT mesenteric lymph nodes and Peyer’s Patches [78]. Importantly, the researchers discovered that therapeutic treatment with probiotics in the first month post-injury resulted in a more anti-inflammatory GALT profile, increasing numbers of CD4^+^CD25^+^FoxP3^+^ regulatory T cells and CD11c^+^ DCs in the mesenteric lymph nodes (Figure 3) [78]. Another group examined the gut microbiome composition of humans with SCI and found that bacterial phyla that produce the short-chain fatty acid butyrate—which is associated with having strong anti-inflammatory effects—were significantly reduced [148]. While not directly demonstrated, these gut microbiome changes might contribute to a proinflammatory state in SCI persons [148].

Since these groundbreaking studies, there has been an explosion of interest in how gut microbiota are altered after SCI, and how modulation of the gut microbiome might improve both neurological and immunological outcomes for SCI individuals [149,150,151,152,153,154,155,156]. In regard to peripheral immunity, a recent study demonstrated that after a T9 contusive injury in mice, the gut microbiome displayed a reduced *Firmicutes* to *Bacteroidetes* phyla ratio as well as an increase in the phylum *Proteobacteria* that contains Gram-negative bacteria that produce the endotoxin LPS, which is associated with systemic inflammation [157]. In fact, the researchers found that sCD14, a marker of systemic inflammation, was significantly increased at 42 days post-SCI. Interestingly, deletion of *Pde4b* to disrupt the TLR4/TNFα/PDE4B axis mitigated the microbiome imbalance observed after SCI, and resulted in reduced endotoxin and sCD14 serum levels, suggestive of reduced inflammation (Figure 3) [157]. Taken together, these studies strongly suggest that improving gut microbiota health may in turn improve immunological function in SCI persons, which is particularly exciting given the accessibility of the gut for therapeutic intervention.

## 5. Conclusions

Immune changes post-SCI have major implications in the quality of life of SCI individuals as well as their treatment. With disruption of descending CNS input to immune organs as well as secondary complications of SCI contributing to SCI-IDS, individuals with SCI are faced with a constant state of inflammation and increased risk of infection. Promisingly, recent preclinical research indicates a wide range of potential interventions that may be able to improve immune function and reduce the risk of infection. However, whether these effects are replicated after chronic immune dysfunction has already occurred, which populations of immune cells should be targeted, and how this affects immunity to various infection types in persons are all unknown facets of immune modulation post-SCI. Importantly, while there are many gaps in knowledge regarding immune function and modulation after SCI that remain to be filled, potential opportunities to identify effective therapeutics to better immune function will undoubtedly result in improved quality of life for those living with SCI.

## Figures and Tables

**Figure 1 biology-10-00928-f001:**
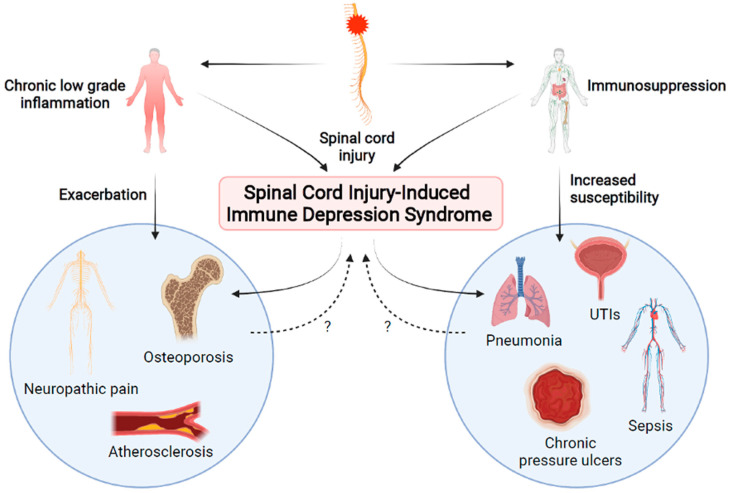
After spinal cord injury (SCI), both chronic systemic inflammation and immunosuppression result in SCI-induced immune depression syndrome (SCI-IDS). Altered immune function results in increased susceptibility to infection and exacerbated secondary complications of SCI. In turn, these secondary complications and infections may create a feedback loop which amplifies immune dysfunction. Created using BioRender.com accessed on 2 September 2021.

**Figure 2 biology-10-00928-f002:**
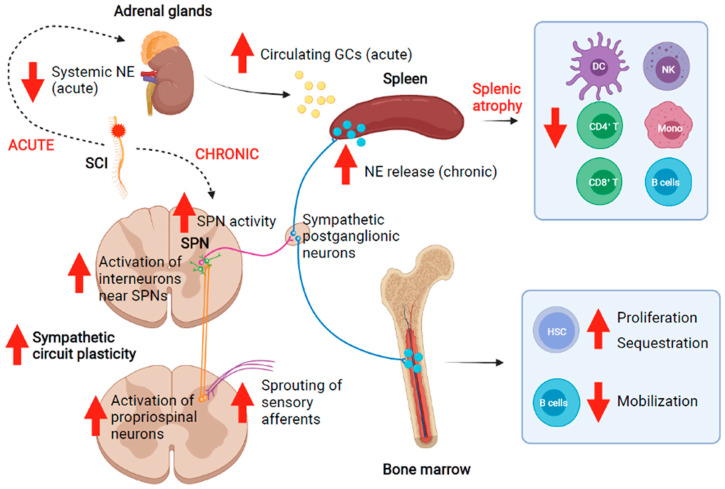
After a SCI, there is an acute drop in systemic norepinephrine (NE) and an increase in plasma glucocorticoids (GCs) released by the adrenal glands. Chronically, increased neural circuit plasticity results in sympathetic hyperreflexia and release of NE by sympathetic post-ganglionic neurons, including those targeting the spleen and other lymphoid organs such as the bone marrow. These changes after SCI result in measurable altered immune profile and function. Created using BioRender.com accessed on 2 September 2021.

**Figure 3 biology-10-00928-f003:**
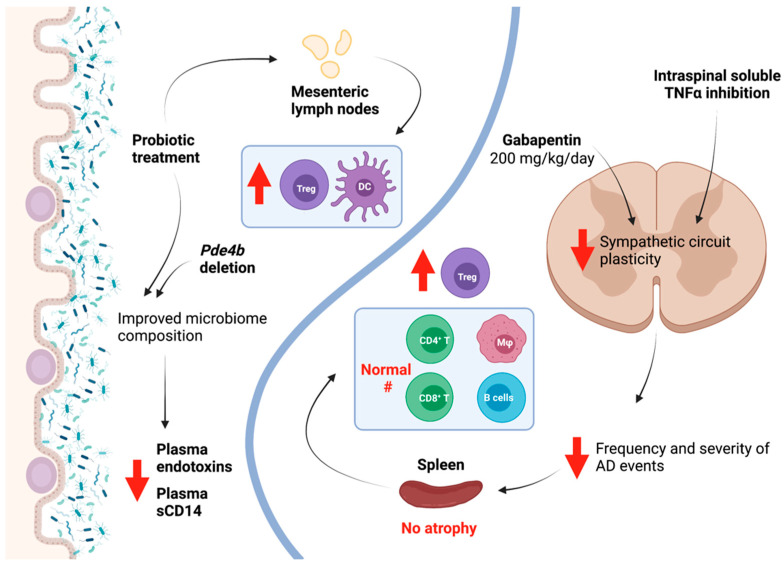
Preclinical studies have revealed that pharmacological inhibition of sympathetic circuit plasticity, either through the use of high-dose gabapentin (GBP) or soluble TNFα inhibition, can reduce autonomic dysreflexia (AD) and improve splenic immune cell profile. Separately, studies have examined the modulation of gut microbiota, either through probiotic treatment or deletion of *Pde4b*, to improve microbiome composition after SCI, which in turn improves anti-inflammatory immune cell profile in gut-associated lymphoid tissues (GALTs), like mesenteric lymph nodes (MLNs), and reduces circulating markers of inflammation. Created using BioRender.com accessed on 2 September 2021.

## Data Availability

Not applicable.

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
