# Peer review of "Peripheral Immune Dysfunction: A Problem of Central Importance after Spinal Cord Injury"

_biology, 2021, doi:10.3390/biology10090928_

Round 1

Reviewer 1 Report

In the manuscript ‘Peripheral Immune Dysfunction: A Problem of Central Importance After Spinal Cord Injury’, authors Jeffries and Tom have discussed immune function in Spinal cord injury (SCI) individuals and potential approaches to improve immunological function.

The authors have neatly summarized the factors that could be responsible for immune dysfunction and the effects of this dysfunction on physiological processes. They review the current available treatment options and why they are insufficient in improving the overall immunity in SCI affected individuals. They further offer alternatives from the current literature that could improve the immune profile of individuals.

Overall this is a very informative review of the current understanding of immune function in individuals after SCI. The review provides useful insights into potential therapeutic interventions that could improve immune function and quality of life of people with SCI.

Minor points:

  1. The review could benefit from addition of schematics. A figure summarizing the current knowledge about the physiological effects post-SCI and drugs used would help the reader grasp multiple concepts easily. This could either be a diagram or a tabulated summary.

Author Response

We thank the Reviewers for their thoughtful review and comments. The Reviewers’ comments are provided below along with a detailed response in blue font. Changes made to the manuscript in response to the comments are tracked in the body of the manuscript.

In the manuscript ‘Peripheral Immune Dysfunction: A Problem of Central Importance After Spinal Cord Injury’, authors Jeffries and Tom have discussed immune function in Spinal cord injury (SCI) individuals and potential approaches to improve immunological function.

The authors have neatly summarized the factors that could be responsible for immune dysfunction and the effects of this dysfunction on physiological processes. They review the current available treatment options and why they are insufficient in improving the overall immunity in SCI affected individuals. They further offer alternatives from the current literature that could improve the immune profile of individuals.

Overall this is a very informative review of the current understanding of immune function in individuals after SCI. The review provides useful insights into potential therapeutic interventions that could improve immune function and quality of life of people with SCI.

Minor points:

  1. The review could benefit from addition of schematics. A figure summarizing the current knowledge about the physiological effects post-SCI and drugs used would help the reader grasp multiple concepts easily. This could either be a diagram or a tabulated summary.

We agree that the addition of schematics would help depict the concepts presented and discussed, and benefit the reader. We now have three figures. Figure 1 is a visualization of the link between the primary SCI and secondary complications and effects, such as increased susceptibility to infection, due to peripheral inflammation and immunosuppression. Figure 2 is a schematic summarizing changes in sympathetic and HPA axis function that result in immunological changes. Finally, Figure 3 presents a visual for the outcomes of current attempts at preventing immune dysfunction.

Reviewer 2 Report

This review on the peripheral immune dysfunctions following SCI is timely and comprehensive in its scope.

The authors however, need to recognize the limited clinical implications of SCI-IDS:

  1. These studies were done in transection animals, and in human SCI, complete transection are exceedingly rare (Harkema et al have shown that spared tissue is always present in cases that were categorized as complete SCIs in humans);
  2. Splenic atrophy has not been proven to be a clinical feature of high-level SCI in humans (https://www.drks.de/drks_web/navigate.do?navigationId=trial.HTML&TRIAL_ID=DRKS00000122, this reviewer is aware of the outcomes of the trial and can say that splenic atrophy was not observed as a clinical feature in these patients); and in pre-clinical models of incomplete injury splenic atrophy was not observed at all (https://www.mdpi.com/1422-0067/20/15/3762) which raises the question of whether a reduced sympathetic tone (rather than a complete abolishment) is sufficient to result in profound peripheral immune deficit;
  3. As high-level SCIs often result in a lack of mobility, a lack of exercise significantly increases the risk of upper respiratory infections (https://www.nature.com/articles/s41598-020-65440-z)

I find that this is a critical element missing from the SCI-IDS literature, and has already begun to dangerously misinform the patient population, who now consider rejecting MPSS-albeit a controversial but tested therapeutic-due to its immunosuppressive effect, as they feel that they are already immune-deficient from a high level SCI.

Author Response

We thank the Reviewers for their thoughtful review and comments. The Reviewers’ comments are provided below along with a detailed response in blue font. Changes made to the manuscript in response to the comments are tracked in the body of the manuscript.

This review on the peripheral immune dysfunctions following SCI is timely and comprehensive in its scope.

The authors however, need to recognize the limited clinical implications of SCI-IDS:

  1. These studies were done in transection animals, and in human SCI, complete transection are exceedingly rare (Harkema et al have shown that spared tissue is always present in cases that were categorized as complete SCIs in humans).

We agree with the Reviewer that a functionally complete injury is rarely anatomically complete. We added discussion on page 7-8 in which we reference the following publications that present data suggesting complete injuries often exhibit tissue sparing: Basso 2000: a review which covers the topic of tissue sparing, even in complete injuries after spinal cord injury; Rejc et al., 2020: a recent paper demonstrating that epidural stimulation elicits motor function recovery in humans with motor complete SCI, indicative of tissue sparing; Kakulas 2004: a review exploring the topic of injury completeness; Ibanez et al., 2021: a recent paper examining motor neuron recruitment order using epidural stimulation to improve functional motor recovery after motor complete injury in humans; Angeli et al., 2018: a clinical paper revealing that combined locomotor training and epidural stimulation in motor complete SCI individuals can result in successful over-ground walking, again suggestive of spared tissue.

We also agree that most of the preclinical studies cited in the review use complete transection models. As noted in the reference mentioned by the Reviewer in their point #2 (Hong et al., 2019), immune dysfunction can occur after incomplete SCI outside of complete transection and splenic atrophy. Moreover, other studies we had cited (Bracchi-Ricard et al., 2016; Kigerl et al., 2016; Norden et al., 2018) made use of incomplete models, sometimes below T6, and also found immunological changes. This suggests that severity and level of injury can impact the extent of immune dysfunction. This important point has now been clarified in the review on page 7-8.

  1. Splenic atrophy has not been proven to be a clinical feature of high-level SCI in humans (https://www.drks.de/drks_web/navigate.do?navigationId=trial.HTML&TRIAL_ID=DRKS00000122, this reviewer is aware of the outcomes of the trial and can say that splenic atrophy was not observed as a clinical feature in these patients); and in pre-clinical models of incomplete injury splenic atrophy was not observed at all (https://www.mdpi.com/1422-0067/20/15/3762) which raises the question of whether a reduced sympathetic tone (rather than a complete abolishment) is sufficient to result in profound peripheral immune deficit.

The reviewer raises a good point about how splenic atrophy is not always observed after SCI. They also raise the important point that altered sympathetic tone may be sufficient to produce a profound immune deficit. Indeed, this is a theme we tried to convey throughout the review. In new text located on page 7-8, we now cite two references (Hong et al., 2018; Hong et al., 2019) which indicate that even though incomplete mid-level thoracic SCI does not result in splenic atrophy, it does result in immune changes. Part of this may be attributed to the SCI being at the level of sympathetic preganglionic neurons that modulate sympathetic input to the spleen, which would fit with the working hypothesis that altered sympathetic tone is a driver of immune dysfunction. We also now include a statement on our personal unpublished observations indicating that splenic immune cell profile changes occur prior to observable splenic atrophy in rats after a T3 complete transection. Additionally, we better emphasize that other preclinical studies making use of incomplete SCI models also observed immune dysfunction, as well as the human studies indicating changes in systemic inflammation after SCI. Lastly, we underscore that injury level and severity may contribute to the extent of immune dysfunction but this is not absolute and the level dependence of SCI-induced immune dysfunction is likely complex. 

  1. As high-level SCIs often result in a lack of mobility, a lack of exercise significantly increases the risk of upper respiratory infections (https://www.nature.com/articles/s41598-020-65440-z).

We agree that this is another risk factor for SCI persons and have added a statement on page 3 including this reference as well as a few others in support of this statement (Ikeda et al., 2021: a clinical study indicating that daily walking exercise reduces pneumonia-related mortality in the aged population; Rice et al., 2020: a clinical study that found reduced walking time by pneumonia patients during hospital stays resulted in higher mortality rates; Ukawa et al., 2019: a clinical study showing that daily walking of at least 1 hour/day is effective in reducing the number of pneumonia-related deaths; Williams 2015: a clinical study which found that walking or running decreases pneumonia-related mortality in a dose-dependent manner).

I find that this is a critical element missing from the SCI-IDS literature, and has already begun to dangerously misinform the patient population, who now consider rejecting MPSS-albeit a controversial but tested therapeutic-due to its immunosuppressive effect, as they feel that they are already immune-deficient from a high level SCI.

As we tried to convey in this review, SCI-IDS is complicated and much more research needs to be conducted to better understand mechanisms that underlie it.

Reviewer 3 Report

The authors have thoughtfully presented the importance of immune response in SCI and how important it is to target it for developing new treatment strategies for the patients. However, a few suggestions and queries mentioned below, if addressed, can improve the overall relevance of the article.

  1. A brief description of SCI in the introduction is required before starting with associated immune dysfunctions.
  2. The monotony of text can be broken down by providing a comprehensive diagrammatic depiction of immune response post SCI.
  3. What are the current therapies for SCI, irrespective of immune response and how targeting immune response for therapeutic interventions be more advantageous than the existing treatments?
  4. A table mentioning the limitations of drugs/compounds used successfully in preclinical settings but that did not work at clinical levels would be informative and easy to understand the status of SCI therapy targeting immune dysfunctions.

Author Response

We thank the Reviewers for their thoughtful review and comments. The Reviewers’ comments are provided below along with a detailed response in blue font. Changes made to the manuscript in response to the comments are tracked in the body of the manuscript.

The authors have thoughtfully presented the importance of immune response in SCI and how important it is to target it for developing new treatment strategies for the patients. However, a few suggestions and queries mentioned below, if addressed, can improve the overall relevance of the article.

  1. A brief description of SCI in the introduction is required before starting with associated immune dysfunctions.

We agree that a straightforward description of spinal cord injury helps to introduce the topic of immune dysfunction to the reader. We have added several sentences to the beginning of the introduction.

  1. The monotony of text can be broken down by providing a comprehensive diagrammatic depiction of immune response post SCI.

We have now added three figures into the manuscript. Figure 1 depicts the overall changes in the body after SCI in relation to immune response. Figure 2 provides a visual of how changes in sympathetic function and the HPA axis result in immune profile changes after SCI.

  1. What are the current therapies for SCI, irrespective of immune response and how targeting immune response for therapeutic interventions be more advantageous than the existing treatments?

We have added more detail into section 4.1, which describes the need for therapies targeting the immune system.

We have expanded slightly on methylprednisolone and the mixed data that exist (Cheung et al., 2015: a review covering the use of methylprednisolone in clinical trials and the mixed data and opinions on its use in SCI persons; Liu et al., 2019: a meta-analysis indicating that methylprednisolone treatment does not provide measurable motor or sensory recovery benefit in the human SCI population, but does increase incidence of respiratory infections and gastrointestinal hemorrhage; Chikuda et al., 2012: a paper revealing increased prevalence of side effects such as gastrointestinal hemorrhage after methylprednisolone treatment in cervical SCI persons; Williams 2018: an informational review on the pharmacology of corticosteroids and their immunological effects).

We have also added information on exercise after SCI as a means to improve inflammation, (Buccholz et al., 2009: a clinical study showing that increased activity time in SCI persons is correlated with lower levels of CRP; Rosety-Rodriguez et al., 2014: a clinical study we have cited earlier in the review indicating that an arm-cranking exercise regimen reduces circulating levels of TNFalpha and IL-6).

While already discussed briefly in our review, we have expanded slightly on current treatments used in both the clinic and clinical/preclinical studies such as antihypertensive medications and epidural stimulation (new additions – Harkema et al., 2018; Harman et al., 2021; West et al., 2015), adding a statement referring to the fact that we do not know how these treatments might affect immune function, though they may do so.

It is generally difficult to say which therapeutic interventions might be more advantageous, because this would depend largely on the desired outcome or measure assessed. For example, when looking at motor recovery, Torres-Espin et al., 2018 (https://academic.oup.com/brain/article/141/7/1946/5025689?login=true) and Schmidt et al., 2021 (https://www.sciencedirect.com/science/article/pii/S0889159120324569) indicated that a low level of systemic inflammation using LPS can improve motor function after SCI. However, how this might be affecting peripheral immune function was not examined, and chronic low-grade inflammation has been linked to a wide range of secondary complications after SCI such as osteoporosis (in Schmidt et al., 2021 – specifically linked to anxiety-like behaviors). Conversely, a treatment targeting immune function could have other side effects that are unintended or undesirable on other facets of recovery and function. Therefore, the treatment that is “best” is likely to be a combination of therapies aimed at targeting multiple facets of the SCI.

  1. A table mentioning the limitations of drugs/compounds used successfully in preclinical settings but that did not work at clinical levels would be informative and easy to understand the status of SCI therapy targeting immune dysfunctions.

To our knowledge, no clinical trials have attempted to examine peripheral systemic immune function as an endpoint. There are some studies on methylprednisolone and minocycline with some interest in modulating inflammation after SCI as a neuroprotective mechanism. However, these have been recently reviewed nicely in Bloom et al., 2020 (https://www.sciencedirect.com/science/article/pii/S0014488619302900) and including these studies would be beyond the scope of our review which is specifically focused on peripheral immune dysfunction.

Reviewer 4 Report

The authors wrote a very interesting review regarding an understudied subject, the peripheral immune dysfunction wich follows SCI. 

The review is well written and is composed of four different chapters which resume perfectly the subject. The references used are constituted both from clinical and experimental studies.

The review is easy to read and understand even for non immunologist.

I do not have any remark or subjection to improve it. 

Author Response

We thank the Reviewers for their thoughtful review and comments. The Reviewers’ comments are provided below along with a detailed response in blue font. Changes made to the manuscript in response to the comments are tracked in the body of the manuscript.

The authors wrote a very interesting review regarding an understudied subject, the peripheral immune dysfunction wich follows SCI. 

The review is well written and is composed of four different chapters which resume perfectly the subject. The references used are constituted both from clinical and experimental studies.

The review is easy to read and understand even for non immunologist.

I do not have any remark or subjection to improve it. 

We thank the reviewer for their positive remarks.

Round 2

Reviewer 2 Report

Figure 1 has issues with font, kindly correct prior to publication.